# Angiotensin II promotes podocyte injury by activating Arf6-Erk1/2-Nox4 signaling pathway

**Guanghua Che**[1], **Hang Gao**[2], **Qibo Hu**[1], **Hongchang Xie**[1], **Yunfeng Zhang**[1]*

**1** Department of Pediatrics, Second Hospital, Jilin University, Changchun, China, **2** The Key Laboratory of Pathobiology, Ministry of Education, Norman Bethune College of Medicine, Jilin University, Changchun, China

\* yunfengzhang11@126.com

**Data Availability Statement:** All relevant data are within the paper and its Supporting Information files.

**Funding:** The author(s) received no specific funding for this work.

## Abstract

Angiotensin II (Ang II) is a key contributor to glomerular disease by predominantly resulting in podocyte injury, whereas the underlying molecular mechanisms has not been fully understood. This study aimed to investigate if and how ADP-ribosylation factor 6 (Arf6), a small GTP-binding protein, involves Ang II-induced cellular injury in cultured human podocytes. Cellular injury was evaluated with caspase 3 activity, reactive oxygen species (ROS) level and TUNEL assay. Arf6 activity was measured using an Arf6-GTP Pull-Down Assay. Ang II significantly enhanced Arf6 expressions accompanied by increase of Arf6-GTP. The TUNEL-positive cells as well as activated caspase 3, NADPH oxidase 4 protein (Nox4) and ROS levels were dramatically increased in Ang II-treated podocytes, which was prevented by secinH3, an Arf6 activity inhibitor. Induction of ROS by Ang II was inhibited in podocytes with Nox4 knockdown. Ang II-induced elevation of Nox4 and ROS was prevented by Arf6 knockdown. Phpspho-Erk1/2$^{Thr202/Tyr204}$ levels were upregulated remarkably following Ang II treatment, and Erk inhibitor LY3214996 significantly downregulated Nox4 expression. In addition, Ang II decreased CD2AP expression. Overexpression of CD2AP prevented Ang II-induced upregulation of Arf6-GTP. Our data demonstrated that Ang II promotes ROS production and podocytes injury through activation of Arf6-Erk1/2-Nox4 signaling. We also provided evidence that Ang II activates Arf6 by degradation of CD2AP.

## Introduction

Glomerular filtration barrier is composed of endothelial cells, glomerular basement membrane and the slit diaphragm (SD) between foot processes (FPs) of podocytes [1]. Podocytes are highly specialized and ultimately differentiated glomerular visceral epithelial cells, playing a critical role in the pathogenesis of proteinuria [2]. Several podocyte proteins, such as nephrin, podocin, CD2AP, α-actinin-4, and transient receptor potential cation channel subfamily C member 6 (TRPC6), have been identified and proved to play important role in maintaining normal podocyte function and glomerular filtration barrier [1,2]. Proteinuria, one of the main clinical manifestations in children with nephrotic syndrome, is mainly caused by podocyte injury which can present with disorganization of actin cytoskeleton, FP effacement, loss of the SD, detachment from glomerular basement membrane and cellular apoptosis or death [1,2].

**Competing interests:** The authors have declared that no competing interests exist.

It has been well known that as the primary effector of the renin–angiotensin system, angiotensin II (Ang II) participates in cellular pathological processes such as inflammation, apoptosis, and fibrosis [3]. Recent studies have shown that Ang II can activate and/or trigger various cellular events (e.g. oxidative stress and reactive oxygen species overproduction, endoplasmic reticulum stress, autophagy and mitochondrial dysfunction), thus resulting in cytoskeletal rearrangement and podocyte apoptosis [4–9]. Notably, NADPH oxidase (Nox) plays a critical role in driving reactive oxygen species (ROS) production. Moreover, it was found that Nox4 is the most abundant isoform of Nox proteins in podocytes and mediates ROS-related podocyte injury in diabetic nephropathy [10]. Nevertheless, the underlying molecular mechanisms by which Ang II causes podocyte injury has not been fully understood.

Just recently, it has been reported that the small GTPase ADP-ribosylation factor 6 (Arf6) belonging to the Ras superfamily is present in glomerular podocytes, and elevation of Arf6 activity is required for focal adhesion turnover and lamellipodia formation following induction of nephrin tyrosine phosphorylation *in vitro* and *in vivo* [11]. The role of Arf protein is controlled by guanine exchange factors (GEFs) and GTPase activating proteins (GAPs), respectively. The GEFs activate Arf6 by catalyzing the exchange of GDP to GTP, whereas the GAPs inactivate Arf6 by promoting GTP hydrolysis [12]. Since our understanding of the role of Arf6 in podocyte injury remains incomplete, we investigated here if and how Arf6 involves Ang II-induced ROS production and cellular apoptosis in cultured human podocytes.

## Materials and methods

### Antibodies

The primary antibodies used in this study are as the below: rabbit anti-Arf6 (cat. no. ab226389; western blot (Wb): 1 : 500; immunofluorescence (IF) staining: 1 : 200; Abcam, Cambridge, MA, USA), mouse anti-β-actin (cat. no. a5441; Wb: 1 : 5,000; Sigma-Aldrich, St. Louis, MO, USA), rabbit anti-Nox4 (cat. no. ab109225; Wb: 1 : 600; IF: 1 : 150; Abcam), mouse anti-phospho-Erk1/$^{2Thr202/Tyr204}$ (cat. no. 9101; Wb: 1 : 1,000; Cell Signaling Technology, Danvers, MA, USA), rabbit anti-Erk1/2 (cat. no. 4695; Wb: 1 : 1000; Cell Signaling Technology), rabbit anti-cleaved caspase-3 antibody (cat. no. ab2302; IF: 1 : 100; Abcam), and rabbit anti-CD2AP (cat. no. ab231320; Wb: 1 : 400; Abcam).

### Human podocytes culture, treatment and transfections

Conditionally immortalized human podocytes, a gift from Dr. Moin Saleem (University of Bristol, Bristol, United Kingdom), were maintained and cultured as described previously [13]. Briefly, cells were cultured in RPMI 1640 medium (Invitrogen, Carlsbad, CA, USA) supplemented with 10% fetal calf serum (Invitrogen), 1x Insulin-Transferrin-Selenium (Gibco, Gaithersburg, MD, USA) and 1% Pen/Strep (Invitrogen). In this study, all studies were performed on human podocyte cell line in passages 6–10.

Cultured human podocytes were treated with Ang II (Enzo Life Sciences, Farmingdale, NY, USA) as indicated concentrations and time duration in the context. In experiments using Ang II receptor antagonists Losartan (Merck Pharmaceuticals, Elkhorn, NE, USA), caspase inhibitor z-VAD-fmk (Sigma-Aldrich), Arf6 inhibitor secinH3 (Sigma-Aldrich), and Erk inhibitor LY3214996 (Sigma-Aldrich), cells were pretreated for 1 h with the indicated inhibitor followed by Ang II administration in the presence of inhibitor.

To knockdown expression of Nox4, human podocytes were transfected with pSilencer 2.1-U6 puro (ThermoFisher Scientific, Wilmington, DE, USA) containing Nox4-siRNA (5'-agccagucaccaucauuucuu-3') and Lipofectamine 2000 (Invitrogen) according to the manufacturer's protocols. The sequence for the control-siRNA (5'-uaaggcuaugaagagauac-3') does not

match any human genes. Stable human Nox4 knockdown podocyte cell line was created by addition of puromycin (final concentration 2.5 μg/ml; Sigma-Aldrich). After two weeks puromycin-resistant cells were collected and western blot assay was performed to verify the knockdown efficiency of Nox4.

Transient knockdown of Arf6 was performed in human podocytes using the prepackaged Arf6 Mission shRNA Lentiviral Transduction Particles (TRCN0000048005, Sigma-Aldrich) and the control shRNA that does not match any human genes, respectively. Human podocytes (1 x $10^6$ cells) were seeded in 6-well plate in the presence of 15 μl of lentiviral particles ($10^6$ TU). After 12 h, Ang II was added at the final concentration of 1 μM for 48 h. Expression of Arf6 and Nox4 as well as ROS level were determined, respectively.

To overexpress CD2AP, pcDNA3.1-CD2AP (human CD2AP mRNA Reference Sequence: NM_012120.3) was cloned, and transiently transfected into podocytes with Lipofectamine 2000 (Invitrogen) according to the manufacturer's protocols. The blank vector pcDNA3.1 (a gift from Oskar Laur, Addgene plasmid # 128034) was used as the controls. After 48 h, cells were collected and analyzed.

## Detection of activated caspase 3 level

The level of activated caspase 3 was measured with the Caspase 3 Colorimetric Protease Assay kit (Invitrogen) according to the manufacturer's protocols. The level of active caspase 3 was expressed as the value of $OD_{405nm}$. Notably, two blank wells without addition of lysates were used as the background. Background absorbance was subtracted from the absorbance of both induced and the uninduced samples.

## TUNEL assay

Apoptotic cell death was assessed with an In-Situ Cell Death Detection Kit, Fluorescein (Sigma-Aldrich) according to the manufacturer's procedure. Nuclei were stained with the DAPI. The TUNEL-positive apoptotic cell nuclei appeared as green under an immunofluorescence microscope (Zeiss, Beijing, China). The percentage of apoptotic cells was calculated and compared.

## Intracellular ROS detection

Podocytes were treated as the indicated, and the intracellular ROS level was measured with the peroxide-sensitive fluorescent probe 2',7'-dichlorodihydrofluorescin diacetate (DCFDA) according to the instructions of DCFDA Cellular ROS Detection Assay (Abcam). DCF fluorescence was detected at excitation and emission wavelengths of 488 and 520 nm, respectively, in a microplate fluorescence reader (BioTek, Beijing, China). The fold change relative to the controls was presented and compared.

## Real time RT-PCR

Total cellular RNA was extracted with TRIzol reagent (Invitrogen) from treated podocytes. Totally, 2 μg of RNA was reversely transcribed into cDNA with the IScript cDNA Synthesis Kit (BioRad, Hercules, CA, USA) following manufacturer's protocol. Real time quantitative PCR (qPCR) was performed for evaluation of Arf6 mRNA level. The PCR reaction were 1x SYBR Green PCR Master Mix (Bio-Rad), 1.5 μl of cDNA and 0.2 μM of Arf6 primers (forward: 5'- atggggaaggtgctatccaaa-3', reverse: 5'-gcagtccactacgaagatgagacc-3', 270 bp). Glyceraldehyde 3-phosphate dehydrogenase (GAPDH) was used as the internal and loading control (forward: 5'-accacagtccatgccatcac-3, reverse: 5'-tccaccac cctgttgctgta-3', 452 bp). The amplification was carried out by an initial denaturation at

95˚C for 5 min followed by 40 cycles of 95˚C for 30 sec, 60˚C for 30 sec and 72˚C for 30 sec. The mRNA level of Arf6 relative to GAPDH was calculated by using the $2^{-\Delta\Delta Ct}$, and the fold-change was presented and compared.

## Western blot assay

Total protein was extracted with the RIPA Lysis Buffer (Millipore Sigma) containing EDTA-free protease inhibitor cocktail (Roche, Shanghai, China). Protein concentration was quantitated using a Bicinchoninic Acid Protein Assay kit (Pierce; Thermo Fisher Scientific). A total of 75 μg protein was separated using 7.5 or 12.5% SDS-PAGE, and transferred to a nitrocellulose membrane (Abcam). Non-specific binding was blocked for 1 h in 5% fat-free milk. The indicated primary antibodies were added and incubated for overnight at 4˚C. After 3 times washes, the membranes were incubated with HRP-conjugated goat anti-rabbit or goat anti-mouse IgG antibodies (Invitrogen) for 1 h. Restore™ Western Blot Stripping Buffer (Thermo Fisher Scientific) was used to remove the primary antibodies for re-probing the other primary antibody. The signals were detected using an ECL Substrate (Thermo Fisher Scientific), and the intensity of the specific bands was quantified using ImageJ software (version 1.51s; NIH, Bethesda, MD, USA).

## Immunofluorescence staining

Podocytes were cultured on cover slides and treated as the indicated. Cells were fixed for 10 min in 4% paraformaldehyde solution followed by a 10 min-permeabilization in 0.1% Triton X-100. Thereafter, 10% goat serum was applied for reducing non-specific staining. Cells were then incubated with rabbit anti-Arf6 or rabbit anti-Nox4 antibody for overnight in cold room, followed by 1 h-incubation with Alexa Fluor 488-conjugated goat anti-rabbit IgG (Invitrogen). Secondary antibody only was used as the blank control. The cover slides were mounted with ProLong Gold Antifade Media with DAPI (Invitrogen), and images were taken on an immunofluorescence microscope (Zeiss).

## Active Arf6 detection

Arf6 activity was determined with the Arf6-GTP Pull-Down Kit following manufacturer's instructions (Cytoskeleton, Denver, CO, USA) and the protocol described previously [11]. Briefly, 500 μg of cell lysates were incubated in 4˚C cold room on a rotator for 1 h with 15 μl of GGA3-PBD bead, then centrifuged at 3,000g at 4˚C for 2 min. GTPγS (final concentration 200 μM) and GDP (final concentration 1 mM) treated samples were used as the positive and negative controls, respectively. Beads were washed 3 times with 600 μl each of Wash Buffer, re-spun, and then mixed with 20 μl of 2x Laemmli sample buffer and thoroughly resuspended the beads by gently tapping the bottom of the tube. The beads were boiled for 2 min, and western blot assay was performed for Arf6 activity detection.

## Statistical analysis

Data are presented as the mean ± standard deviation (S.D.). One-way ANOVA with the Turkey's post-hoc test was used for statistical analysis (GraphPad Prism6.0, GraphPad Software, Inc., La Jolla, CA, USA). $P < 0.05$ was considered to have significant difference.

## Results

### Arf6 involves Ang II-induced podocyte injury

We firstly explored podocyte injury by assessing the caspase 3 activation level in Ang II-treated podocytes. Activated caspase 3 is an important indicator for evaluation of cellular apoptosis

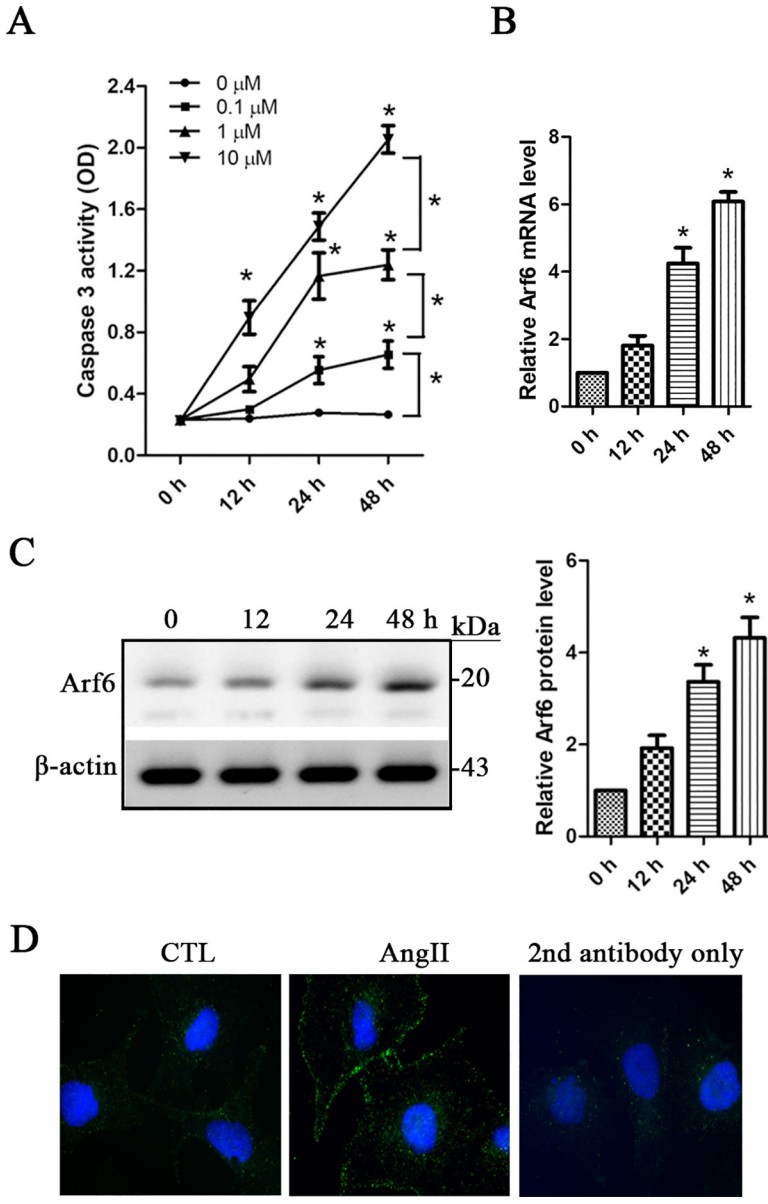

**Fig 1. Arf6 is related to Ang II-induced human podocyte injury. A.** Human podocytes were treated with angiotensin II (Ang II) as indicated. Apoptosis was evaluated with caspase 3 activity. Data are presented as mean ± SD. n = 5. $^*p < 0.01$ *vs. 0 h or as indicated comparison.* **B** and **C.** Human podocytes were treated with 1 μM of Ang II for 12, 24 and 48 h, respectively. The mRNA (**B**) and protein (**C**) levels of Arf6 was assessed and compared. Data are presented as mean ± SD. n = 3. $^*p < 0.01$ *vs. 0 h. Full length blots are provided in S1 Fig.* **D.** Human podocytes were treated with 1 μM of Ang II for 24 h, and indirect immunofluorescence was performed for Arf6 staining (green color). The non-treated cells were used as the controls (CTL). The 2nd antibody only was used for the blank control. *Magnification x 40.*

[14]. Our data showed a dose- and time-dependent increase of caspase 3 activity following Ang II treatment in human podocytes (**Fig 1A**). We then evaluated Arf6 expression levels in Ang II-treated podocytes. After 24 h, both the mRNA and protein levels of Arf6 were signifi-cantly upregulated in 1μM of Ang II-treated podocytes (**Fig 1B and 1C**). The distribution of Arf6 was also assessed using indirect immunofluorescence staining in Ang II-treated podo-cytes. An increased signal of Arf6 particularly along cell plasma membrane was revealed at 48

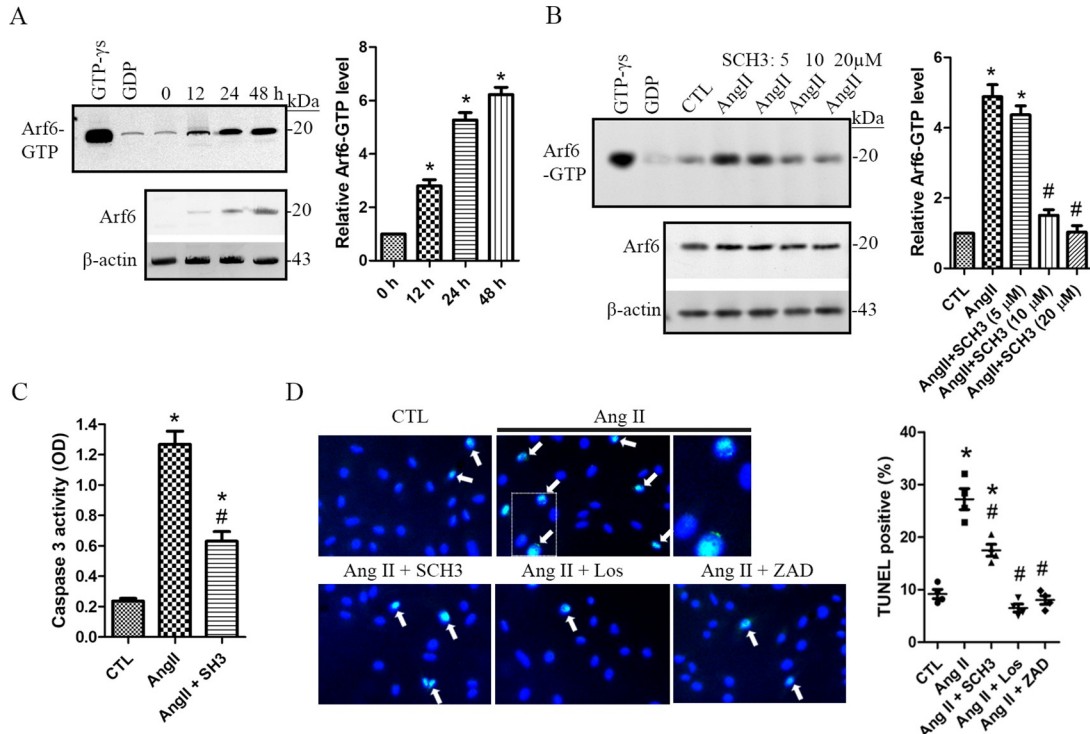

**Fig 2. Arf6 involves Ang II-induced human podocyte injury. A.** Human podocytes were treated with 1 μM of angiotensin II (Ang II) for 12, 24 and 48 h, respectively. The levels of Arf6-GTP and total Arf6 were analyzed and compared. GTP-γs and GDP treated samples were used as the positive and negative control, respectively. Data are presented as mean ± SD. n = 3. $^*p < 0.01$ vs. *0 h.* **B** and **C.** Human podocytes were treated with Ang II for 24 h in the absence or presence of secinH3 (SH3). The non-treated cell was used as the controls (CTL). The levels of Arf6-GTP and total Arf6 were analyzed and compared **(B)**. The effect of 10 μM of secinH3 was evaluated on caspase 3 activity in Ang II-treated podocytes **(C)**. Data are presented as mean ± SD. n = 5. $^*p < 0.001$ vs. CTL, $^\#p < 0.05$ vs. Ang II. **D.** Human podocytes were treated with Ang II for 24 h in the absence or presence of secinH3 (10 μM), losartan (1 μM) and v-ZAD-fmk (10 μM). The vehicle DMSO treated cells were used as the controls (CTL). TUNEL assay was performed for evaluation of apoptotic cell death (arrows). Data are presented as mean ± SD. n = 4. $^*p < 0.01$ vs. CTL, $^\#p < 0.01$ vs. Ang II. Magnification x 10. Full length blots of A and B are provided in S2 Fig.

h following 1μM of Ang II treatment (**Fig 1D**). These findings suggest that Arf6 may be related to Ang II-induced podocyte apoptosis. Arf6 is a small GTP-binding protein, we thus investigated the effect of Ang II on its activation level. The Arf6-GTP pull-down assay showed that the levels of GTP-bound Arf6 were significantly enhanced at 12 h persisting to 48 h following 1 μM of Ang II treatment (**Fig 2A**), indicating a time-dependent activation of Arf6. SecinH3 acts as a Sec7 domain-binding selective antagonist against a GEF protein cytohesin [15]. In this study, secinH3 was used as an Arf protein activation inhibitor as described previously [15]. We found that secinH3 remarkably decreased the levels of active Arf6-GTP in a dose-dependent manner in 1 μM of Ang II-treated podocytes (**Fig 2B**). Moreover, 10 μM of secinH3 dramatically inhibited Ang II-induced increase of caspase 3 activity (**Fig 2C**). Furthermore, we assessed cellular apoptosis with the TUNEL assay. The percentage of apoptotic cell death was significantly higher (Ang II: 27.2 ± 3.98 vs CTL: 9.1 ± 1.82, *p < 0.01*) in 1 μM of Ang II-treated podocytes than that in the controls (**Fig 2D**). Our data also indicated that Ang II-induced apoptosis was significantly prevented by an Ang II receptor antagonist losartan (6.5 ± 1.57), Arf6 inhibitor secinH3 (17.5 ± 2.29) and caspase 3 activation inhibitor z-VAD-fmk (8.0 ± 1.65) (**Fig 2D**). These findings demonstrated that Arf6 activation-mediated caspase 3 signaling is related to Ang II-induced apoptosis in human podocytes.

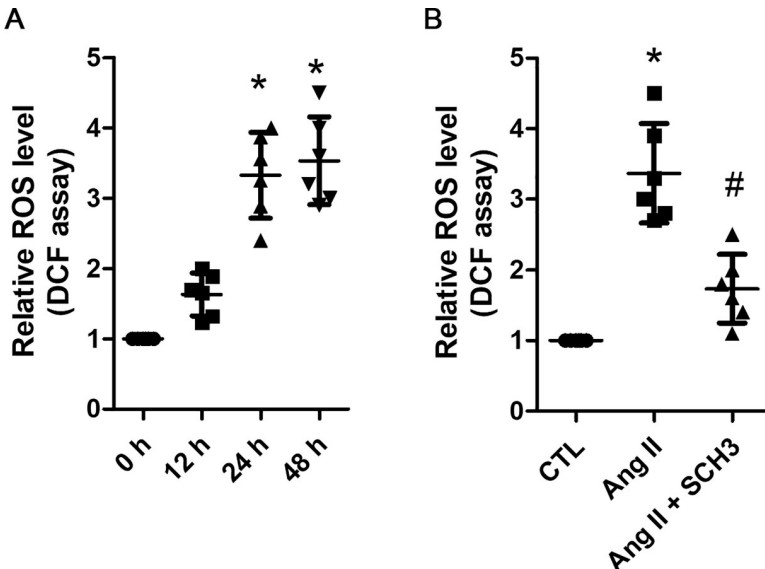

**Fig 3. Ang II increases ROS production through Arf6 in human podocytes. A.** Human podocytes were treated with 1 μM of angiotensin II (Ang II) for 12, 24 and 48 h, respectively. The ROS levels were measured using DCF assay and compared. Data are presented as mean ± SD. n = 6. $^*p < 0.01$ vs. 0 h. **B.** Human podocytes were treated with 1 μM of Ang II for 24 h in the absence or presence of 10 μM of secinH3 (SCH3). The non-treated cell was used as the controls (CTL). The ROS levels were measured and compared. Data are presented as mean ± SD. n = 6. $^*p < 0.01$ vs. CTL, $^#p < 0.05$ vs. Ang II.

Intracellular ROS plays a crucial role in induction of cellular damage [16]. In our study, increase of ROS levels was detected at both 24 and 48 h after 1 μM of Ang II treatment (**Fig 3A**), which was significantly prevented by the application of secinH3 (**Fig 3B**). Therefore, these data indicated that Ang II induces podocyte injury most likely by upregulation of Arf6 activity.

## Active Arf6 mediates upregulation of Nox4 through activation of Erk1/2 signaling in human podocytes

The Nox family proteins have been recognized as one of the main sources of intracellular ROS in a variety of human cell types [17]. It has been reported that upregulation of Nox4 plays an important role in renal oxidative stress and kidney injury [18]. In this study, we found that 1 μM of Ang II significantly increased Nox4 expression in a time-dependent manner (**Fig 4A**). To clarify the role of Nox4 in ROS production, we established stable human podocyte cell line expressing the siNox4, in which Nox4 protein level was significantly knocked down compared to the cells expressing the control siRNAs (**Fig 4B**). Ang II-induced ROS production was dramatically inhibited in the podocytes stably expressing the siNox4 (**Fig 4C**), indicating that Ang II increases ROS production through upregulation of Nox4.

To establish the potential relationship between activated Arf6 and Nox4, we investigated Nox4 expression levels in Ang II-treated podocytes with lentiviral shArf6. Our data showed that shArf6 but not the control shRNA significantly prevented increase of Nox4 protein and ROS levels in 1 μM of Ang II-treated cells (**Fig 4D and 4E**). We also assessed the Nox4 expression levels in Ang II -treated podocytes in the presence or absence of secinH3. As expected, we found that secinH3 significantly inhibited Ang II-induced increase of Nox4 (**Fig 5A**). Indirect immunofluorescence staining with anti-Nox4 antibody also showed that increased signal of Nox4 by Ang II was obviously prevented in the presence of secinH3 (**Fig 5B**). Therefore, these findings demonstrated that Arf6 indeed mediates Ang II-induced Nox4 upregulation in human podocytes.

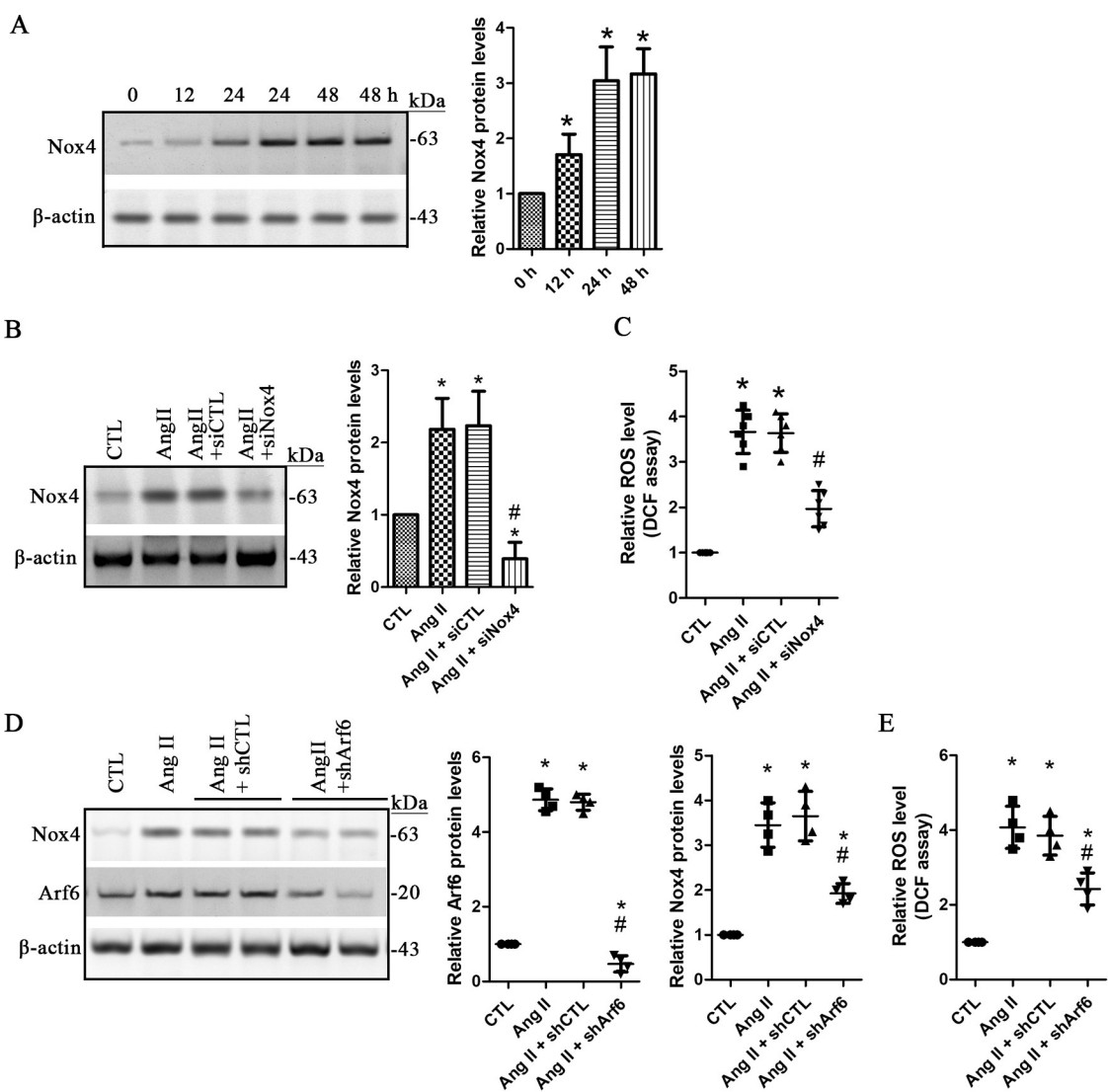

**Fig 4. Ang II increases ROS production through Nox4 in human podocyte. A.** Human podocytes were treated with 1 μM of angiotensin II (Ang II) for 12, 24, and 48 h, and the protein levels of Nox4 were assessed and compared. Data are presented as mean ± SD. n = 3. $^*p < 0.01$ *vs. 0 h.* **B** and **C.** Podocytes stably expressing siRNA-Nox4 (siNox4) or siRNA-control (siCTL) were treated with 1 μM of Ang II for 24 h. The non-treated podocytes were used as the controls (CTL). The protein levels of Nox4 **(B)** and the ROS **(C)** were assessed and compared. Data are presented as mean ± SD. n = 3 (B) and 6 (C). $^*p < 0.01$ *vs. CTL,* $^\#p < 0.05$ *vs. Ang II or Ang II + siCTL.* **D** and **E.** Podocytes were transduced with 15 μl of lentiviral shRNA-Arf6 (shArf6) and the control shRNA (shCTL), respectively. The non-treated cells were used as the controls (CTL). The protein levels of Nox4 and Arf6 **(D)** and the ROS level **(E)** were assessed. Data are presented as mean ± SD. n = 3 (D) and 4 (E). $^*p < 0.01$ *vs. CTL,* $^\#p < 0.05$ *vs. Ang II or Ang II + shCTL. Full length blots of A, B and D are provided in S3 Fig.*

It has been reported that extracellular signal–regulated kinases (Erk1/2) play a critical role in regulation of Nox4 expression [19]. We then explored the activation level of Erk1/2 at Thr202 and Tyr204 following 1 μM of Ang II treatment. Our data showed that Erk1/2 was rapidly activated at 5 min and persistent to 24 h and 48 h following Ang II treatment (**Fig 5C**). We also found that inhibition of Erk1/2 signaling by LY3214996 significantly decreased Nox4 expression level in 1 μM of Ang II-treated podocytes (**Fig 5D**). These data suggest that Ang II upregulates Arf6-GTP activity, which then activates Erk1/2 signal being responsible for induction of Nox4 expression.

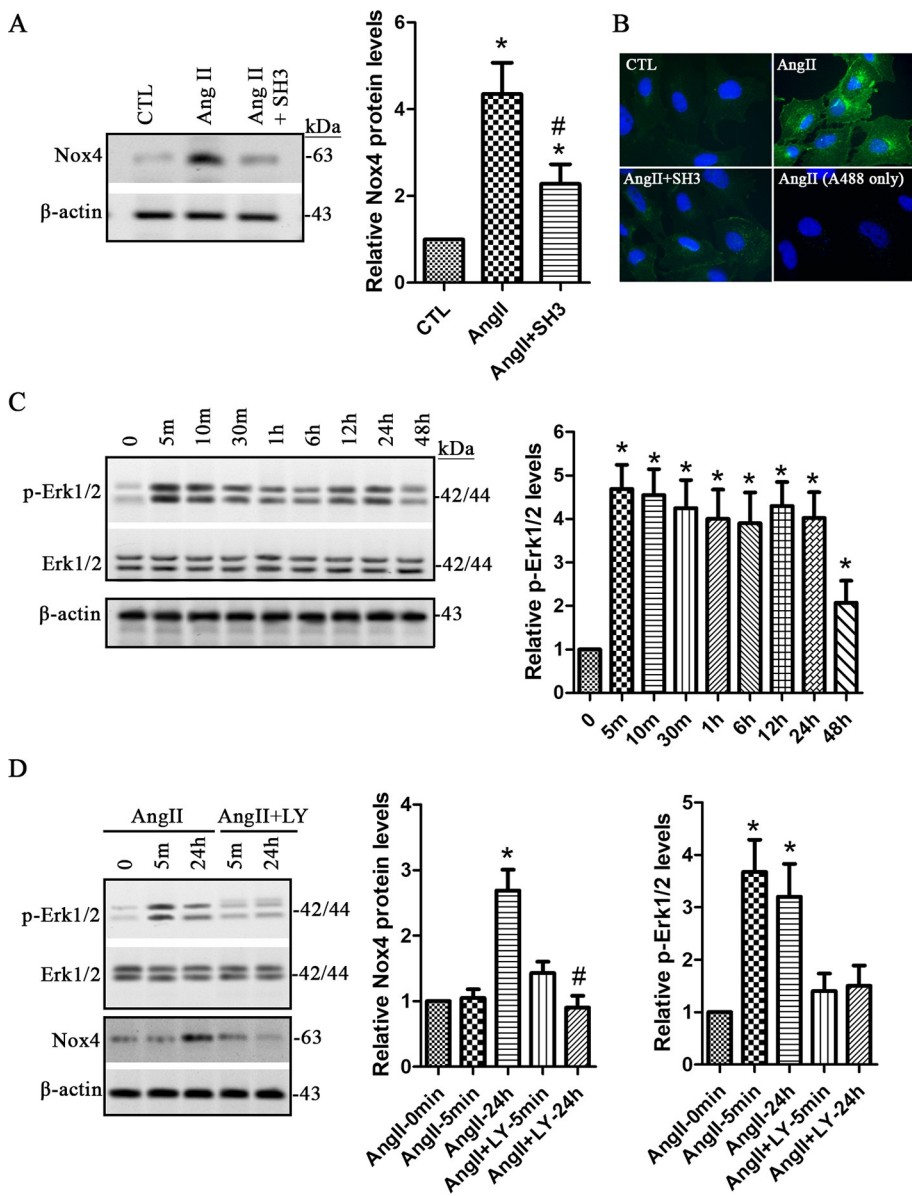

**Fig 5. Increased Arf6-GTP elevates Nox4 through Erk1/2 in Ang II-treated human podocyte.** Human podocytes were treated with 1 μM of angiotensin II (Ang II) for 24 h in the absence or presence of 10 μM of secinH3 (SH3). **A.** The levels of Nox4 were evaluated and compared. Data are presented as mean ± SD. n = 3. *p < 0.01 vs. CTL, #p < 0.05 vs. Ang II. **B.** Indirect immunofluorescence staining was performed for Nox4 (green color). The secondary antibody Alexa 488-conjugated goat anti-rabbit antibody alone was used for the blank control (A488 only). *Magnification x 20.* **C.** The activation level of Erk1/2 was assessed using immunoblot assay. Data are presented as mean ± SD. n = 3. *p < 0.01 vs. 0 h. **D.** Human podocytes were pretreated for 1 h with the Erk1/2 inhibitor LY3214996 (1 μM; LY), and then Ang II was added at the final concentration of 1 μM and incubated for 24 h in the presence of 1 μM of LY3214996. The protein levels of Nox4 were evaluated. Data are presented as mean ± SD. n = 3. *p < 0.01 vs. AngII-0min, #p < 0.01 vs. Ang II-24h. Full length blots of A, C and D are provided in S4 Fig.

## Reduction of CD2AP is responsible for Arf6-GTP activation in Ang II-treated human podocytes

It was reported that in MDCK cells, overexpression of ectopic Cindr, the CD2-associated protein (CD2AP) form in *Drosophila*, inhibited activation of Arf6 [20]. In this study, we found

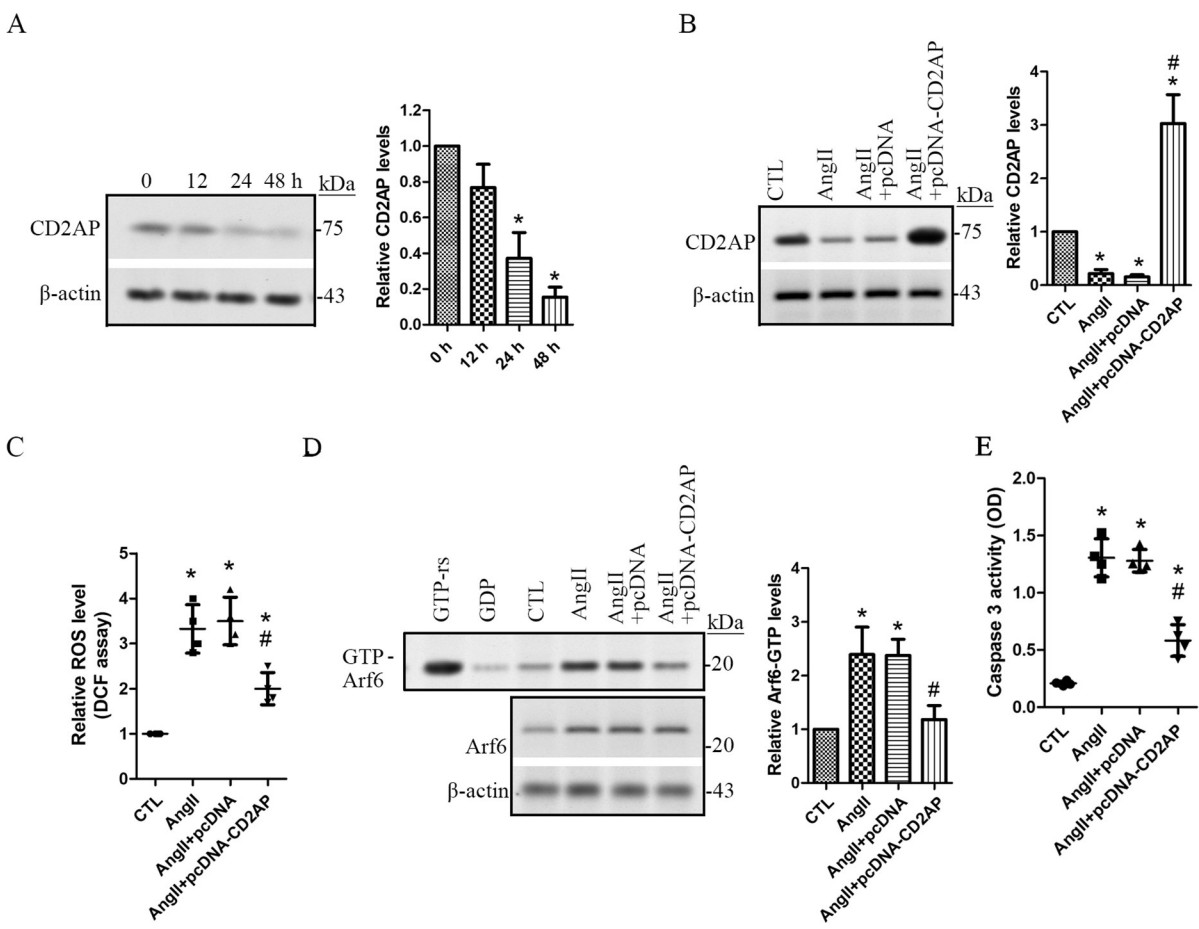

**Fig 6. Ang II increases Arf6 activity through downregulation of CD2AP.** Human podocytes were treated with 1 μM of angiotensin II (Ang II) for 12, 24 and 48 h. **A.** The protein levels of CD2AP were assessed using immunoblot assay. Data are presented as mean ± SD. n = 3. $^*p < 0.01$ *vs. 0 h.* **B-E.** Human podocytes were transfected with pcDNA3.1CD2AP and pcDNA3.1 blank vector, respectively. 24 h later, 1 μM of Ang II was added and incubated for 24 h. The expressions of CD2AP (**B**) and the levels of ROS (**C**), Arf6-GTP (**D**), and apoptosis (**E**) were evaluated and compared. Data are presented as mean ± SD. n = 3 (B and D) and 4 (C and E). $^*p < 0.05$ *vs. CTL,* $^\#p < 0.05$ *vs. Ang II or Ang II + BV. Full length blots of A, B and D are provided in S5 Fig.*

that CD2AP expression was significantly decreased following 1 μM of Ang II treatment (**Fig 6A**). We then tested if rescue of CD2AP expression can reduce active Arf6-GTP level in Ang II-treated podocytes. We transfected human podocytes with pcDNA3.1-CD2AP to overexpress human CD2AP. Our data showed that overexpression of CD2AP was obtained in the pcDNA3.1-CD2AP transfected podocytes, but not the blank vector pcDNA3.1 alone (**Fig 6B**). In human podocytes overexpressing CD2AP, Arf6 activation level was significantly decreased following Ang II treatment (**Fig 6C**). Moreover, Ang II-mediated increases of the ROS and caspase 3 activity were also prevented by CD2AP overexpression (**Fig 6D and 6E**). These data suggest that Ang II decreases CD2AP expression which is responsible for upregulation of Arf6 activity in human podocytes.

## Discussion

Damage to podocytes plays a crucial role in the development of proteinuria and kidney disease [1,2]. Ang II mediates podocytes injury directly by alteration of expression and distribution of podocyte proteins [21], and indirectly by induction of cellular hypertrophy, ROS and cellular

apoptosis [8]. In this study, elevation of ROS production and caspase 3 activity as well as the percentage of apoptotic cell death were detected following Ang II treatment in human podocytes. Losartan, an Ang II type 1 receptor (AT1R) antagonist, significantly inhibited Ang II induced podocyte apoptosis, suggesting a specific role of Ang II in induction of podocyte apoptosis. A large amount of *in vitro* studies has shown that Ang II results in podocyte apoptosis [4–6, 22,23]. Although some studies also show that Ang II can cause podocyte apoptosis in *in vivo* model [22], it should be noted that there is no reliable evidence that Ang II directly induce *in vivo* podocyte apoptosis. The biological significance of apoptosis need be further investigated in Ang II-induced podocyte injury. Nox protein is a major inducer of oxidative stress and ROS production in a variety of cell types [17]. Nox4 as the most abundant isoform of Nox proteins in podocytes plays multiple roles in the presence of different stimuli [10]. Elevation of transforming growth factor (TGF)-β in kidney tissue is related to podocyte damage such as induction of apoptosis and detachment. In mouse podocytes treated with TGF-β1, upregulation of Nox4 increases ROS production and cellular apoptosis [24]. High glucose-stimulated Nox4 activation induces apoptosis in cultured mouse podocytes as well as in diabetic mouse models [25,26]. In the present study, our data showed that Ang II-induced increase of ROS production was significantly prevented in human podocytes with stable knockdown of Nox4, suggesting that upregulation of Nox4 is responsible for Ang II-induced increase of ROS production.

The small GTPase Arf6 is involved in membrane trafficking and cell motility. Arf6 operates by cycling between the GDP-bound (inactive) and the GTP-bound (active) forms [12]. Nephrin, an essential podocyte protein, is a critical component of podocyte slit diaphragm protein complex [27]. Nephrin tyrosine phosphorylation triggers cytoskeletal dynamics associated with increased lamellipodia activity and focal adhesion turnover [11,28]. It has been demonstrated that Arf6 mediates nephrin tyrosine phosphorylation-induced podocyte focal adhesion remodeling and lamellipodia formation [11]. In addition, it has been reported that overexpression of dominant-negative Arf6$^{T27N}$ completely inhibits VEGF-induced Rac1 activation and ROS production in cultured endothelial cells [29]. AIP1, a novel GTPase-activating protein (GAP) for Arf6 that binds to the SH3 domain of cytosolic subunit p47phox *via* its proline-rich region, can disrupt formation of an active Nox2 complex, attenuating ROS production in human endothelial cells [30]. In cultured vascular smooth muscle cells, Ang II promotes activation of Arf6 to control ROS production by regulation of Nox1 expression [31]. We then tested if Arf6 also involves Ang II-induced ROS production in human podocyte. Our data showed that both expressions and activation of Arf6 were significantly elevated following Ang II treatment in human podocytes. Arf6 activation by the cytohesin family belonging to Arf GEFs is inhibited by secinH3, a cell permeable triazole compound [15]. In our study, inhibition of Arf6 activity by both secinH3 and Arf6 knockdown dramatically prevented Ang II-induced caspase 3 activation as well as Nox4 and ROS production, suggesting that elevation of active Arf6-GTP by Ang II is required for Nox4 upregulation and ROS production. Just recently, it has been reported that in dynamin1/2-deficient primary podocytes, Ang II induces abnormal membrane dynamics with increased Rac1 activation and lamellipodial extension, which was attenuated in deficiency of AT1R [32]. This finding suggests that the internalization of AT1R is blunted and Ang II signal is prolonged in podocytes with dynamin1/2 deficiency. Nephrin tyrosine phosphorylation augments Rac1 activity through Arf6, thus leading to focal adhesion turnover and lamellipodial formation in cultured human podocytes [11]. In addition, it has been reported that in cultured podocytes, Ang II treatment results in actin cytoskeleton reorganization, cell adhesion reduction, actin-associated protein downregulation, and albumin permeability increase, in which TRPC6-mediated decrease of MYH9 plays a crucial role [33]. Therefore, actin cytoskeleton remodeling, lamellipodial activity, cell adhesion alteration, and

permeability change are the important mechanisms by which Ang II induces structural and functional podocyte injury and thus filtration barrier disruption. The limitation of our study is that we only used apoptosis as the index of podocyte injury. However, our findings reveal a novel function of the small GTPase Arf6 in the context of Ang II induced podocyte injury. Effects of Arf6 activation on actin cytoskeleton, lamellipodial formation, cell adhesion, and permeability should be further evaluated in Ang II treated podocytes.

The Erks are widely expressed intracellular protein kinases that are involved in regulation of multiple cellular events [34]. Erks are also known to activate many transcription factors (e.g. c-myc, c-Fos, AP-1, and Elk1) responsible for regulation of various target genes [35]. It has been found that Ang II stimulates Erk1/2 activation and Nox4-derived ROS production in glomerular mesangial cells [36]. In the current study, the phospho-Erk1/2$^{Thr202/Tyr204}$ levels were rapidly increased following Ang II treatment. Notably, inhibition of Erk1/2 signaling by LY3214996 significantly decreased Ang II-mediated upregulation of Nox4. These data suggest that Ang II elevates Nox4 expression through activation of Erk1/2 signaling pathway in human podocytes.

The adaptor protein, CD2AP, is initially identified as a T-cell adaptor protein. CD2AP is also localized at the podocyte slit diaphragm, and plays a crucial role in maintaining normal podocyte function [37]. In rat kangaroo kidney epithelial cells (PtK1 cell line) expressing constitutively activated Arf6$^{Q67L}$, CD2AP was enriched at one end of F-actin tails [38]. In MDCK cells, overexpression of ectopic Cindr, a CD2AP form in *Drosophila*, significantly suppressed active Arf6-GTP [20]. These findings suggest a direct or indirect role of CD2AP on regulation of Arf6 activity. Here, reduction of CD2AP was detected following Ang II treatment, and overexpression of CD2AP significantly inhibited Ang II-induced elevation of active Arf6-GTP, indicating that CD2AP plays an important role in suppression of Arf6 activation in human podocytes. We further demonstrated that overexpression of CD2AP prevented Ang II-induced caspase 3 activation and ROS production. However, the precise role and molecular mechanisms by which reduction of CD2AP enhanced Arf6 activation in Ang II-treated podocytes are still not clear. We propose that CD2AP may inhibit Arf6 activation possibly by binding to Arf-GAPs proteins; but this issue should be further investigated in proteinuric kidney disease *in vitro* and *in vivo*.

## Conclusions

Taken together, our data demonstrated that Ang II promotes ROS production and apoptosis through activation of Arf6-Erk1/2-Nox4 signaling, in which reduction of CD2AP is responsible for Arf6 activation. These findings suggest that rescue of CD2AP expression or target against Arf6 activation may be a novel potential way for alleviating podocyte injury in such situations with abundant Ang II.

## Supporting information

**S1 Checklist. PLOS ONE clinical studies checklist.**
(DOCX)

**S1 Fig. Full length blots of Fig 1C.**
(PDF)

**S2 Fig. Full length blots of Fig 2A and 2B.**
(PDF)

**S3 Fig. Full length blots of Fig 4A, 4B and 4D.**
(PDF)

**S4 Fig. Full length blots of** Fig 5A, 5C and 5D.
(PDF)

**S5 Fig. Full length blots of** Fig 6A, 6B and 6D.
(PDF)

## Author Contributions

**Conceptualization:** Guanghua Che, Yunfeng Zhang.

**Data curation:** Guanghua Che, Hang Gao, Qibo Hu, Hongchang Xie.

**Formal analysis:** Guanghua Che, Hang Gao, Qibo Hu.

**Investigation:** Guanghua Che, Hang Gao, Qibo Hu.

**Methodology:** Guanghua Che, Hang Gao, Qibo Hu, Hongchang Xie.

**Project administration:** Yunfeng Zhang.

**Software:** Hang Gao, Hongchang Xie.

**Supervision:** Yunfeng Zhang.

**Validation:** Guanghua Che, Hang Gao, Hongchang Xie.

**Writing – original draft:** Guanghua Che.

**Writing – review & editing:** Guanghua Che, Hang Gao, Yunfeng Zhang.

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
