## [Decision Letter · Decision Letter 0]

10 Jan 2020

PONE-D-19-34582

Angiotensin II promotes podocyte injury by activating Arf6-Erk1/2-Nox4 signaling pathway

PLOS ONE

Dear Dr. Zhang,

Thank you for submitting your manuscript to PLOS ONE. After careful consideration, we feel that it has merit but does not fully meet PLOS ONE’s publication criteria as it currently stands. Therefore, we invite you to submit a revised version of the manuscript that addresses the points raised during the review process.

We would appreciate receiving your revised manuscript by Feb 24 2020 11:59PM. To enhance the reproducibility of your results, we recommend that if applicable you deposit your laboratory protocols in protocols.io, where a protocol can be assigned its own identifier (DOI) such that it can be cited independently in the future. For instructions see: http://journals.plos.org/plosone/s/submission-guidelines#loc-laboratory-protocols

We look forward to receiving your revised manuscript.

Kind regards,

Michael Bader

Academic Editor

PLOS ONE

Journal Requirements:

2.  We noticed you have some minor occurrence(s) of overlapping text with the following previous publication(s), which needs to be addressed:

https://doi.org/10.1152/ajprenal.00438.2013

https://doi.org/10.1371/journal.pone.0184575

https://doi.org/10.1159/000453163

https://doi.org/10.3892/mmr.2018.9637

In your revision ensure you cite all your sources (including your own works), and quote or rephrase any duplicated text outside the Methods section. Further consideration is dependent on these concerns being addressed.

Reviewers' comments:

Reviewer's Responses to Questions

**Comments to the Author**

1. Is the manuscript technically sound, and do the data support the conclusions?

Reviewer #1: Partly

Reviewer #2: Yes

2. Has the statistical analysis been performed appropriately and rigorously? 

Reviewer #1: Yes

Reviewer #2: Yes

3. Have the authors made all data underlying the findings in their manuscript fully available?

Reviewer #1: Yes

Reviewer #2: Yes

4. Is the manuscript presented in an intelligible fashion and written in standard English?

Reviewer #1: Yes

Reviewer #2: Yes

5. Review Comments to the Author

Reviewer #1: In this paper, the authors showed that Ang II increases, via suppression of CD2AP, Arf6 mRNA RNA, protein, and activity in cultured human podocytes. Afr6 enhances Nox4 and ROS production, and then activates caspase 3. Although precise mechanisms underlining Arf6 activation by Ang II were not shown, all data consistently support the authors’ proposal. The major concern is that Ang II does not induce cell death in podocytes in vivo. Moreover, apoptosis is hardly ever observed in podocytes in vivo.

1. The authors should show that the number of dead cells after Ang II stimulation. How much % of podocytes are actually dead dependently on caspase 3. As generally recommended (for example, PMID:17562483), apoptosis should be demonstrated by multiple methods including TUNEL staining. The authors should show whether the effect of Ang II on cell death is blocked by AT1 antagonist and caspase 3 inhibitor.

2. Active caspase3 is generally stained at perinuclear region, but in Figure 2D, it is overlapped with DAPI staining. There is a concern about specificity of the staining.

3. In Fig1D, the staining observed in areas apart from DAPI staining. The authors should present the corresponding phase contrast image to show the location of the Arf6 and morphological change of the cells.

4. Ang II does not induce apoptosis in podocyte in vivo. The authors should discuss about this discrepancy and biological significance of their findings.

5. The precise role of CD2AP in activation of Arf6 by Ang II is not clear. Possible molecular mechanism should be discussed.

6. In the abstract, “mall” should be “small”.

7. P3, l54, “Angiotensin (Ang II)” should be “Angiotensin II (Ang II)” or “Angiotensin (Ang) II”

8. The supplier of Ang II should be shown.

9. P9, l192, “in the presence of absence of SecinH3” should be “in the presence or absence of SecinH3”

10. “BV” is an uncommon abbreviation.

Reviewer #2: In the present manuscript “Angiotensin II promotes podocyte injury by activating Arf6-Erk1/2-Nox4 signaling Pathway”，the authors，Che et al. demonstrated the role of Arf6 in Ang II-induced podocyte injury and shown that Ang II promotes ROS production and apoptosis through activation of Arf6-Erk1/2-Nox4 signaling, in which reduction of CD2AP is responsible for Arf6 activation. This is a well-written paper containing interesting results which merit publication. For the benefit of the reader, however, the authors should consider addressing a few minor concerns.

1.The authors have used Arf6 inhibitor in their studies to show its role of podocyte injury. However, inhibitor sometimes not so specific because they work by affecting gene expression or activity. Thus, a set of graphs to show the expression of downstream genes by knocking down Arf6 is needed urgently.

2.In Figure 1D, the enlarged picture is somewhat fuzzy. Please change the pictures with high resolution image.

3.Figure 4, there is something excess in the panel of β-actin in 4A and Nox4 in 4B. Please confirm the exposure time and treatment groups in the images.

4.Many bar charts were used in this paper. It is kindly suggested to replace it with scatter plot with bar graphs to better represent the number and the dispersion degree of specimens.

5.The manuscript is very well written and easy to understand; however, it can use some minor grammatical corrections to improve the flow.

6. PLOS authors have the option to publish the peer review history of their article (what does this mean?). If published, this will include your full peer review and any attached files.

Reviewer #1: No

Reviewer #2: No

---

## [Author Response · Author response to Decision Letter 0]

27 Jan 2020

We would like to thank the reviewer for reviewing our manuscript and their comments. Point by point responses are outlined below. 

Reviewer #1: 

In this paper, the authors showed that Ang II increases, via suppression of CD2AP, Arf6 mRNA RNA, protein, and activity in cultured human podocytes. Afr6 enhances Nox4 and ROS production, and then activates caspase 3. Although precise mechanisms underlining Arf6 activation by Ang II were not shown, all data consistently support the authors’ proposal. The major concern is that Ang II does not induce cell death in podocytes in vivo. Moreover, apoptosis is hardly ever observed in podocytes in vivo.

1. The authors should show that the number of dead cells after Ang II stimulation. How much % of podocytes are actually dead dependently on caspase 3. As generally recommended (for example, PMID:17562483), apoptosis should be demonstrated by multiple methods including TUNEL staining. The authors should show whether the effect of Ang II on cell death is blocked by AT1 antagonist and caspase 3 inhibitor.

Response: As you recommended, the results from TUNEL assay have been provided in the revised manuscript (Fig 2D), showing that the percentage of apoptotic cell death was increased significantly in podocytes treated with Ang II. The data from AT1 antagonist (Losartan) and caspase 3 inhibitor (z-VAD-fmk) were also included, showing that the administration of Losartan and ZAD-fmk significantly reduced Ang II-induced apoptosis.

2. Active caspase3 is generally stained at perinuclear region, but in Figure 2D, it is overlapped with DAPI staining. There is a concern about specificity of the staining.

Response: As you recommended, we replaced the active caspase 3 staining with the images from TUNEL assay in the revised manuscript (Fig 2D). 

3. In Fig1D, the staining observed in areas apart from DAPI staining. The authors should present the corresponding phase contrast image to show the location of the Arf6 and morphological change of the cells.

Response: Arf6 is known to be involved in vesicle trafficking and cellular morphology. It has been shown that Arf6 is also localized both in the cytoplasm and in the cell plasma membrane in podocyte (Lin JS, et al. PLoS ONE. 2017. 12(9): e0184575.), which supports our Arf6 staining pattern. We did not take the phase contrast images; but in the revised manuscript, we provided the blank control staining image (the 2nd antibody only was applied while staining was performed) to show specific staining for Arf6. In addition, the anti-Arf6 antibody that we used is commercial. Therefore, the staining of Arf6 should be specific.

4. Ang II does not induce apoptosis in podocyte in vivo. The authors should discuss about this discrepancy and biological significance of their findings.

Response: Podocytes are ultimately differentiated glomerular visceral epithelial cells. In proteinuric kidney disease, severe injury results in podocyte differentiation, cell death, detachment, and loss of podocytes. Podocyte loss plays a key role in the development of proteinuria and kidney disease. Ang II executes hemodynamic effects on renal tissue, and has a direct influence on induction of podocyte injury by altered expression and distribution of podocyte proteins. Ang II can also promote podocyte injury indirectly by inducing cellular hypertrophy and apoptosis. Some studies have shown that Ang II can result in podocyte apoptosis both in vitro and in vivo (Márquez E, et al. Renin-angiotensin system within the diabetic podocyte. Am J Physiol Renal Physiol. 2015;308(1):F1-10; Gao Z, et al. Dab1 Contributes to Angiotensin II-Induced Apoptosis via p38 Signaling Pathway in Podocytes. Biomed Res Int. 2017;2017:2484303; Jia J, Ding G, Zhu J, Chen C, Liang W, Franki N, Singhal PC. Angiotensin II infusion induces nephrin expression changes and podocyte apoptosis. Am J Nephrol. 2008;28(3):500-7; Ren Z, Liang W, Chen C, Yang H, Singhal PC, Ding G. Angiotensin II induces nephrin dephosphorylation and podocyte injury: role of caveolin-1. Cell Signal. 2012;24(2):443-50; Chen X, et al. c-Abl mediates angiotensin II-induced apoptosis in podocytes. J Mol Histol. 2013;44(5):597-608.). However, the role of apoptotic podocytes in the pathogenesis of proteinuric kidney disease should be fully investigated in vivo. Although increasing evidences have been provided showing Ang II can induce podocyte apoptosis through Nox4, this study adds the small GTPase Arf6 to this key cellular event as a novel signaling mechanism. We added this key point and references in the revised manuscript.

5. The precise role of CD2AP in activation of Arf6 by Ang II is not clear. Possible molecular mechanism should be discussed.

Response: Yes, the precise role and molecular mechanisms by which reduction of CD2AP enhanced Arf6 activation in Ang II-treated podocytes are still not clear. We propose that CD2AP may inhibit Arf6 activation possibly by binding to ArfGAPs proteins; but this issue should be further investigated in vitro and in vivo. We discussed this possible mechanism in the revised manuscript.

6. In the abstract, “mall” should be “small”.

Response: We are really sorry for the typing error. We have corrected it, and also checked the entire manuscript carefully for spelling and grammar.

7. P3, l54, “Angiotensin (Ang II)” should be “Angiotensin II (Ang II)” or “Angiotensin (Ang) 

II”

Response: We have corrected it. Thank you.

8. The supplier of Ang II should be shown.

Response: The supplier of Ang II was added in the revised manuscript.

9. P9, l192, “in the presence of absence of SecinH3” should be “in the presence or absence of 

SecinH3”

Response: We are really sorry for the typing error. We have corrected it in the revised version.

10. “BV” is an uncommon abbreviation.

Response: We avoided use of BV, and use the “pcDNA” for blank vector.

Reviewer #2: 

In the present manuscript “Angiotensin II promotes podocyte injury by activating Arf6-Erk1/2-Nox4 signaling Pathway”, the authors, Che et al. demonstrated the role of Arf6 in Ang II-induced podocyte injury and shown that Ang II promotes ROS production and apoptosis through activation of Arf6-Erk1/2-Nox4 signaling, in which reduction of CD2AP is responsible for Arf6 activation. This is a well-written paper containing interesting results which merit publication. For the benefit of the reader, however, the authors should consider addressing a few 

minor concerns.

1.The authors have used Arf6 inhibitor in their studies to show its role of podocyte injury. However, inhibitor sometimes not so specific because they work by affecting gene expression or activity. Thus, a set of graphs to show the expression of downstream genes by knocking down Arf6 is needed urgently.

Response: Thank you for your recommendations. We performed knockdown assay with lentiviral shRNA-Arf6 and assessed Nox4 expression and ROS level. These data have been provided in the revised manuscript (Fig 4D,E). 

2.In Figure 1D, the enlarged picture is somewhat fuzzy. Please change the pictures with high resolution image.

Response: High resolution image has been provided for Fig 1D.

3.Figure 4, there is something excess in the panel of ß-actin in 4A and Nox4 in 4B. Please confirm the exposure time and treatment groups in the images.

Response: For unknown reason, sometimes, smear bands were observed in some lanes, likely resulted by an appropriate storage of lysates, unenough boiling, or the lysis buffer itself. Anyway, we replaced it with the samples from the other independent experiment.

4.Many bar charts were used in this paper. It is kindly suggested to replace it with scatter plot with bar graphs to better represent the number and the dispersion degree of specimens.

Response: Thank you for your suggestions, we provided the scatter plot images for (Fig 2D, Fig 3, Fig 4C-E, and Fig 6C,E) in the revised manuscript.

5.The manuscript is very well written and easy to understand; however, it can use some minor grammatical corrections to improve the flow.

Response: The revised manuscript has been carefully checked for spelling and grammar.

We also addressed “Journal Requirements” as the below:

a. We formatted the manuscript according to PLOS ONE’s style requirements.

b. We avoided overlapping text with your indicated publications, and added the references in the revised manuscript.

c. Full length blots of western blot assay are provided as the Supplementary images in Supporting information.

---

## [Decision Letter · Decision Letter 1]

10 Feb 2020

PONE-D-19-34582R1

Angiotensin II promotes podocyte injury by activating Arf6-Erk1/2-Nox4 signaling pathway

PLOS ONE

Dear Dr. Zhang,

Thank you for submitting your manuscript to PLOS ONE. After careful consideration, we feel that it has merit but does not fully meet PLOS ONE’s publication criteria as it currently stands. Therefore, we invite you to submit a revised version of the manuscript that addresses the points still raised by reviewer 1.

We would appreciate receiving your revised manuscript by Mar 26 2020 11:59PM. To enhance the reproducibility of your results, we recommend that if applicable you deposit your laboratory protocols in protocols.io, where a protocol can be assigned its own identifier (DOI) such that it can be cited independently in the future. For instructions see: http://journals.plos.org/plosone/s/submission-guidelines#loc-laboratory-protocols

We look forward to receiving your revised manuscript.

Kind regards,

Michael Bader

Academic Editor

PLOS ONE

Reviewers' comments:

Reviewer's Responses to Questions

**Comments to the Author**

1. If the authors have adequately addressed your comments raised in a previous round of review and you feel that this manuscript is now acceptable for publication, you may indicate that here to bypass the “Comments to the Author” section, enter your conflict of interest statement in the “Confidential to Editor” section, and submit your "Accept" recommendation.

Reviewer #1: (No Response)

Reviewer #2: All comments have been addressed

2. Is the manuscript technically sound, and do the data support the conclusions?

Reviewer #1: Yes

Reviewer #2: Yes

3. Has the statistical analysis been performed appropriately and rigorously? 

Reviewer #1: Yes

Reviewer #2: Yes

4. Have the authors made all data underlying the findings in their manuscript fully available?

Reviewer #1: Yes

Reviewer #2: Yes

5. Is the manuscript presented in an intelligible fashion and written in standard English?

Reviewer #1: Yes

Reviewer #2: Yes

6. Review Comments to the Author

Reviewer #1: 1. % of dead podocytes and TUNEL should be presented

The authors added the data of TUNEL staining and showed dependency on caspase 3. Judging from very high rate of TUNEL even in control cells (~10%), the cells the author used were very prone to apoptosis, probably all will die within a few days.

2. Localization of caspase 3 staining was nuclear.

The authors provided reasonable pictures of caspase 3.

3. Phase contrast pictures would be presented for Fig 1D.

The authors did not provide phase contrast pictures, but this concern is not a central issue.

5. Mechanism of Arf6 inhibition by CD2AP is not clear.

The authors adequately responded by adding a possible mechanism in the discussion.

6-10. Minor errors

The authors adequately amended the errors.

4. Ang II does not induce apoptosis in vivo.

The authors should notice that TUNEL pictures in the reference 23 (Gao Z et al) and 26 (Chen X et al) show many positive staining in tubules without injury nor fragmentation of nuclei, strongly suggesting non-specific staining.　Moreover, the EM pictures of ref 24 and 26 do not show apoptosis. These unreliable data are misleading and should not be cited as evidences for in vivo apoptosis. The authors should cite more recent and reliable papers, such as JASN 30: 2307–2320, 2019. doi: https://doi.org/10.1681/ASN.2019010053. This paper demonstrated that Ang II directly exaggerate injury in podocytes of Dynamin knockout mice, in which internalization of AT1R is impaired, but no apoptosis was detected. Judging from the authors’ Figure 5C data, it is speculated that the internalization of AT1R is blunted and Ang II signal is prolonged in the cell line used.

The authors should clearly state that there is no reliable evidence that Ang II directly induce apoptosis in podocyte in vivo, and discuss about biological significance of their findings.

Reviewer #2: In the present manuscript “Angiotensin II promotes podocyte injury by activating Arf6-Erk1/2-Nox4 signaling Pathway”，the authors，Che et al. demonstrated the role of Arf6 in Ang II-induced podocyte injury and shown that Ang II promotes ROS production and apoptosis through activation of Arf6-Erk1/2-Nox4 signaling, in which reduction of CD2AP is responsible for Arf6 activation. The author has corrected the deficiencies of previous article. This is a well-written paper containing interesting results which merit publication.

7. PLOS authors have the option to publish the peer review history of their article (what does this mean?). If published, this will include your full peer review and any attached files.

Reviewer #1: No

Reviewer #2: No

---

## [Author Response · Author response to Decision Letter 1]

12 Feb 2020

We would like to thank the reviewers for reviewing our manuscript and their comments. Point by point responses are outlined below. 

Reviewer #1: 

1. % of dead podocytes and TUNEL should be presented.

The authors added the data of TUNEL staining and showed dependency on caspase 3. Judging from very high rate of TUNEL even in control cells (~10%), the cells the author used were very prone to apoptosis, probably all will die within a few days.

Response: All studies were performed on human podocyte cell line in passages 6-10. For TUNEL assay, cells were cultured on coverslips in 12-well plate and pretreated for 1 h with secinH3 (10 µM), losartan (1 µM) and v-ZAD-fmk (10 µM), respectively. Thereafter, Ang II was applied for 24 h at the concentration of 1 µM. Nontreated cell was used as the controls, but equal amount of DMSO was added because v-ZAD-fmk and secinH3 was dissolved in DMSO. The application of DMSO may have a subtle influence on cellular apoptosis. In addition, we did not see more apoptotic cells for long period culture (e.g. 72 h) in this condition. To make it clear, we added this information and the data (the number of the percentage of apoptotic cell: CTL: 9.1 ± 1.82, Ang II: 27.2 ± 3.98, Ang II+SCH3: 17.5 ± 2.29, Ang II+Los: 6.5 ± 1.57, Ang II+ ZAD 8.0 ± 1.65) in the revised manuscript. We hope that our responses are suitable. Thank you very much.

2. Localization of caspase 3 staining was nuclear.

The authors provided reasonable pictures of caspase 3.

Response: Thank you.

3. Phase contrast pictures would be presented for Fig 1D.

The authors did not provide phase contrast pictures, but this concern is not a central issue.

Response: Thank you.

5. Mechanism of Arf6 inhibition by CD2AP is not clear.

The authors adequately responded by adding a possible mechanism in the discussion.

Response: Thank you.

6-10. Minor errors 

The authors adequately amended the errors.

Response: Thank you.

4. Ang II does not induce apoptosis in vivo.

The authors should notice that TUNEL pictures in the reference 23 (Gao Z et al) and 26 (Chen X et al) show many positive staining in tubules without injury nor fragmentation of nuclei, strongly suggesting non-specific staining. Moreover, the EM pictures of ref 24 and 26 do not show apoptosis. These unreliable data are misleading and should not be cited as evidences for in vivo apoptosis. The authors should cite more recent and reliable papers, such as JASN 30: 2307–2320, 2019. doi: https://doi.org/10.1681/ASN.2019010053. This paper demonstrated that Ang II directly exaggerate injury in podocytes of Dynamin knockout mice, in which internalization of AT1R is impaired, but no apoptosis was detected. Judging from the authors’ Figure 5C data, it is speculated that the internalization of AT1R is blunted and Ang II signal is prolonged in the cell line used. The authors should clearly state that there is no reliable evidence that Ang II directly induce apoptosis in podocyte in vivo, and discuss about biological significance of their findings.

Response: Thank you for your recommendations. We removed the references 23, 24 and 26 to avoid misunderstanding and misleading to this key point. About this issue, we discussed as the below:

A large amount of in vitro studies has shown that Ang II results in podocyte apoptosis [4-6, 22,23]. Although some studies also show that Ang II can cause podocyte apoptosis in in vivo model [22], it should be noted that there is no reliable evidence that Ang II directly induce in vivo podocyte apoptosis. The biological significance of apoptosis need be further investigated in Ang II-induced podocyte injury. 

Just recently, it has been reported that in dynamin1/2-deficient primary podocytes, Ang II induces abnormal membrane dynamics with increased Rac1 activation and lamellipodial extension, which was attenuated in deficiency of AT1R [32]. This finding suggests that the internalization of AT1R is blunted and Ang II signal is prolonged in podocytes with dynamin1/2 deficiency. Nephrin tyrosine phosphorylation augments Rac1 activity through Arf6, thus leading to focal adhesion turnover and lamellipodial formation in cultured human podocytes [11]. In addition, it has been reported that in cultured podocytes, Ang II treatment results in actin cytoskeleton reorganization, cell adhesion reduction, actin-associated protein downregulation, and albumin permeability increase, in which TRPC6-mediated decrease of MYH9 plays a crucial role [33]. Therefore, actin cytoskeleton remodeling, lamellipodial activity, cell adhesion alteration, and permeability change are the important mechanisms by which Ang II induces structural and functional podocyte injury and thus filtration barrier disruption. The limitation of our study is that we only used apoptosis as the index of podocyte injury. However, our findings reveal a novel function of the small GTPase Arf6 in the context of Ang II induced podocyte injury. Effects of Arf6 activation on actin cytoskeleton, lamellipodial formation, cell adhesion, and permeability should be further evaluated in Ang II treated podocytes.

---

## [Editor Report · Decision Letter 2]

13 Feb 2020

Angiotensin II promotes podocyte injury by activating Arf6-Erk1/2-Nox4 signaling pathway

PONE-D-19-34582R2

Dear Dr. Zhang,

We are pleased to inform you that your manuscript has been judged scientifically suitable for publication and will be formally accepted for publication once it complies with all outstanding technical requirements. Please especially make sure that Fig. 1C is not upside down, the bands look strange.

With kind regards,

Michael Bader

Academic Editor

PLOS ONE
---

## [Editor Report · Acceptance letter]

18 Feb 2020

PONE-D-19-34582R2 

Angiotensin II promotes podocyte injury by activating Arf6-Erk1/2-Nox4 signaling pathway 

Dear Dr. Zhang:

I am pleased to inform you that your manuscript has been deemed suitable for publication in PLOS ONE. Congratulations! Your manuscript is now with our production department. 

With kind regards,

on behalf of

Prof. Michael Bader 

Academic Editor

PLOS ONE